# Histomolecular Validation of [^18^F]-FACBC in Gliomas Using Image-Localized Biopsies

**DOI:** 10.3390/cancers16142581

**Published:** 2024-07-18

**Authors:** Benedikte Emilie Vindstad, Anne Jarstein Skjulsvik, Lars Kjelsberg Pedersen, Erik Magnus Berntsen, Ole Skeidsvoll Solheim, Tor Ingebrigtsen, Ingerid Reinertsen, Håkon Johansen, Live Eikenes, Anna Maria Karlberg

**Affiliations:** 1Department of Circulation and Medical Imaging, Norwegian University of Science and Technology, 7030 Trondheim, Norway; 2Department of Pathology, St. Olavs Hospital, Trondheim University Hospital, 7030 Trondheim, Norway; 3Department of Clinical and Molecular Medicine, Norwegian University of Science and Technology, 7030 Trondheim, Norway; 4Department of Neurosurgery, Ophthalmology and Otorhinolaryngology, University Hospital of North Norway, 9019 Tromsø, Norway; 5Department of Radiology and Nuclear Medicine, St. Olavs Hospital, Trondheim University Hospital, 7030 Trondheim, Norway; 6Department of Neurosurgery, St. Olavs Hospital, Trondheim University Hospital, 7030 Trondheim, Norway; 7Department of Neuroscience, Norwegian University of Science and Technology, 7030 Trondheim, Norway; 8Department of Clinical Medicine, Faculty of Health Sciences, UiT The Arctic University of Norway, 9019 Tromsø, Norway; 9Department of Health Research, SINTEF Digital, 7034 Trondheim, Norway

**Keywords:** [^18^F]-FACBC, glioma, amino acid PET, MRI

## Abstract

**Simple Summary:**

Gliomas are the most common type of malignant brain tumors in adults. They are frequently heterogeneous, containing regions of varying properties and aggressiveness. It can be challenging to define the tumor borders and identify the most aggressive parts of the tumor based on MRI alone. This study investigates whether PET imaging with amino acid tracer [^18^F]-FACBC can provide additional information on the composition of the tumor. The results suggest that uptake of the tracer could be used to identify aggressive tumor tissue with high accuracy, and with higher sensitivity than that of contrast-enhanced MRI.

**Abstract:**

Background: Gliomas have a heterogeneous nature, and identifying the most aggressive parts of the tumor and defining tumor borders are important for histomolecular diagnosis, surgical resection, and radiation therapy planning. This study evaluated [^18^F]-FACBC PET for glioma tissue classification. Methods: Pre-surgical [^18^F]-FACBC PET/MR images were used during surgery and image-localized biopsy sampling in patients with high- and low-grade glioma. TBR was compared to histomolecular results to determine optimal threshold values, sensitivity, specificity, and AUC values for the classification of tumor tissue. Additionally, PET volumes were determined in patients with glioblastoma based on the optimal threshold. [^18^F]-FACBC PET volumes and diagnostic accuracy were compared to ce-T1 MRI. In total, 48 biopsies from 17 patients were analyzed. Results: [^18^F]-FACBC had low uptake in non-glioblastoma tumors, but overall higher sensitivity and specificity for the classification of tumor tissue (0.63 and 0.57) than ce-T1 MRI (0.24 and 0.43). Additionally, [^18^F]-FACBC TBR was an excellent classifier for IDH1-wildtype tumor tissue (AUC: 0.83, 95% CI: 0.71–0.96). In glioblastoma patients, PET tumor volumes were on average eight times larger than ce-T1 MRI volumes and included 87.5% of tumor-positive biopsies compared to 31.5% for ce-T1 MRI. Conclusion: The addition of [^18^F]-FACBC PET to conventional MRI could improve tumor classification and volume delineation.

## 1. Introduction

Gliomas are the most common type of malignant brain tumor in adults and cause significant morbidity and mortality [1]. Accurate classification of the tumor is essential to estimate overall prognoses and select the best treatment strategy for each patient [2]. Gliomas are known to be heterogeneous, sometimes containing regions with different World Health Organization (WHO) Central Nervous System (CNS) tumor grades and molecular properties [3]. Identifying the most aggressive regions of the tumor can be important for histomolecular tissue sampling and surgical resection, along with selecting targets and margins for radiation therapy.

Magnetic resonance imaging (MRI), particularly gadolinium contrast-enhanced T1-weighted (ce-T1) MRI, is widely used for the diagnosis and treatment planning of glioma. However, MRI has limitations when it comes to distinguishing tumor grades, identifying true tumor extension, and differentiating viable tumor tissue from other changes such as edema, inflammation, or radionecrosis. Furthermore, as ce-T1 MRI relies on a compromised blood–brain barrier (BBB) for the delivery of contrast agents, some high-grade tumors do not show contrast enhancement. In contrast to ce-T1, amino acid (AA) positron emission tomography (PET) tracers have been shown to cross the intact BBB and can provide additional diagnostic information regarding the metabolic heterogeneity of the tumor [4]. Therefore, AA PET is recommended by current guidelines as a complement to MRI in glioma diagnostics, resection, biopsy sampling, treatment planning, and therapy response assessment [2,5].

*Anti*-1-amino-3-[^18^F]-fluorocyclobutane-1-carboxylic acid ([^18^F]-FACBC), also known as fluciclovine [^18^F] or Axumin ^®^ (Blue Earth Diagnostics Ltd., Oxford, UK), is an AA PET tracer that has shown preferential glioma cell uptake and low uptake in normal brain parenchyma. A few studies have evaluated the diagnostic performance of [^18^F]-FACBC in gliomas, suggesting benefits in the detection of glioma tissue not detectable with contrast-enhanced MRI [6,7,8,9]. Additionally, higher tumor-to-background ratios (TBRs) were found for [^18^F]-FACBC compared to currently recommended AA PET tracers [^11^C]-MET, [^18^F]-FET, and [^18^F]-FDOPA due to lower uptake in normal brain tissue [10,11]. This difference in TBR could be explained by the fact that [^11^C]-MET, [^18^F]-FET, and [^18^F]-FDOPA are transported mainly via the leucine-preferring transport system L (LAT1), while [^18^F]-FACBC uptake has been found to correspond mainly with the expression of alanine-serine-cysteine transporter 2 (ASCT2) [12,13]. These results highlight [^18^F]-FACBC as another promising candidate for neurooncological PET imaging. However, suitable threshold values for the detection of tumor tissue and tumor delineation have not yet been validated for [^18^F]-FACBC, unlike with the other AA tracers [14,15,16,17,18].

In this study, the diagnostic accuracy of [^18^F]-FACBC PET for the classification of tumor characteristics and composition in heterogenous tumors was evaluated by comparing [^18^F]-FACBC uptake with immunohistomolecular (IHC) and DNA methylation analysis results from image-localized biopsies as a reference standard. The aim of this study was to determine if the addition of [^18^F]-FACBC PET can identify and classify tumor tissue better than ce-T1 MRI alone to help delineate tumor volumes and guide decisions in the planning of biopsy sampling, surgical resection, and radiotherapy for glioma patients in a clinical setting.

## 2. Materials and Methods

### 2.1. Study Design

This prospective study validated the diagnostic accuracy of [^18^F]-FACBC PET (index test) against histomolecular analysis of image-localized biopsies (reference standard). The study design and reporting were guided by the standards for the reporting of diagnostic accuracy studies (STARD) [19].

### 2.2. Study Participants

Adult patients with suspected primary or recurrent glioma (n = 37) were referred to a pre-surgical [^18^F]-FACBC PET/MRI from the neurosurgical departments at St. Olav’s Hospital, Trondheim, and the University Hospital of North Norway, Tromsø, Norway, from November 2019 to June 2021. The patient selection was based on convenience sampling. The inclusion/exclusion process is illustrated in Figure 1. Six patients were excluded due to withdrawn consent (n = 1) or problems with tracer delivery (n = 5). The remaining 31 patients underwent pre-surgical [^18^F]-FACBC PET/MRI examination and, of these, 21 patients were eligible for and consented to sampling up to 4 image-localized biopsies during surgery. Biopsies from 4 patients were excluded from analysis due to loss of image-localization data (n = 2) or uncertain clinical diagnosis of the tumor based on the main biopsy (n = 2). The image-localized biopsies from the remaining 17 patients (9 females, aged 24–74) were included and analyzed in this paper. Exclusion criteria were pregnancy, breastfeeding, pacemakers or defibrillators not compatible with 3 Tesla (T) MRI, no ability to obtain informed consent (e.g., due to severe dysphasia or cognitive deficits), weight > 120 kg, and Karnofsky performance status < 70. All participants gave written informed consent. This study was approved by the Regional Ethics Committee (REK South East Norway, reference number: 2018/2243).

### 2.3. [^18^F]-FACBC PET/MRI Imaging

Two identical hybrid PET/MRI systems (Siemens Biograph mMR, software version Syngo MR VE11P, Erlangen, Germany) were used for simultaneous PET and MRI acquisitions. Patients were injected with [^18^F]-FACBC (3 ± 0.2 MBq/kg) on the scanner examination table at the start of the PET and MRI imaging, and list-mode PET was acquired 0 to 45 min post-injection (p.i.).

MRI sequences were acquired according to current consensus recommendations on standardized brain tumor imaging protocols [20,21]. These included pre- and post-contrast-enhanced 3D T1 magnetization prepared rapid gradient echo imaging (MPRAGE), 3D fluid-attenuated inversion recovery (FLAIR), and T2, as well as an ultrashort echo time (UTE) sequence for PET attenuation correction (AC) purposes.

The last 15 min of the acquisition (30–45 min p.i.) were used for the reconstruction of static PET images, chosen based on previously acquired dynamic PET results for [^18^F]-FACBC in patients with glioma [8]. The images were reconstructed with iterative reconstruction (3D OSEM algorithm, 3 iterations, 21 subsets, 344 × 344 matrix, 4 mm Gaussian filter, voxel size: 2.1 × 2.1 × 2.0 mm^3^) with point spread function correction, decay correction, scatter correction, and AC. AC was based on the UTE sequence and deep learning method DeepUTE developed by Ladefoged et al. [22,23].

### 2.4. Image-Localized Biopsies

Static PET images were fused with FLAIR and ce-T1 MRI prior to surgery using the software PMOD (version 4.304, PMOD Technologies LLC, Zürich, Switzerland). FLAIR, ce-T1, and the fused PET/FLAIR and PET/ce-T1 images were imported to Brainlab (Brainlab AG, Munich, Germany) (n = 14) or SonoWand (Neuraxon AS, Athens, Greece) (n = 3). These images were used for navigation during histomolecular tissue sampling and surgery together with standalone FLAIR and ce-T1 MRI. Additionally, intraoperative ultrasound (US) images were recorded during sampling. These were used to correct for brain shift relative to the preoperative images post-surgery to acquire more precise biopsy localization. This process is explained in further detail in Section 2.6. In total, 1–4 image-localized biopsies were sampled from each patient. Where possible, the biopsies were sampled from both PET signal hotspots and PET-negative areas. All biopsies were taken from areas with increased FLAIR signal.

### 2.5. Histomolecular Analysis

A total of 60 image-localized biopsies were sampled from the 17 included patients. The inclusion process of the image-localized biopsies for immunohistochemical (IHC) and/or DNA methylation analysis and the corresponding results are illustrated in Figure 2 and further details are provided in Appendix A. In short, the samples were subject to DNA methylation, HE staining, and immunohistochemical analysis with IDH1, ATRX, and proliferation indexes (assessed by Ki67). Three specimens were not analyzed due to loss of image-localization data. Of the remaining 57 biopsies, 9 had insufficient material or yielded inconclusive results from both IHC and DNA methylation analysis, leaving 48 biopsies included in this study (IHC analysis: n = 17, DNA methylation analysis: n = 10, IHC and DNA methylation analysis: n = 21).

In total, 48 biopsies were classified as diffuse high-grade glioma (HGG), diffuse low-grade glioma (LGG), or non-tumor tissue, where HGG represents tumor tissue corresponding to CNS WHO grades 3 and 4 and LGG represents tumor tissue corresponding to CNS WHO grade 2. Non-tumor tissue was defined as normal tissue and/or mild inflammatory changes, while necrotic tissue was classified as HGG. This classification was performed by a neuropathologist and based on IHC analysis (n = 17), DNA methylation analysis (n = 10), or both (n = 21). IDH1 mutation status was also determined in the same way. HE staining and IHC analysis determined cell density and ATRX mutation status (n = 38). From DNA methylation analysis, 27 of 31 biopsies were classified into 4 different subtypes (glioblastoma, astrocytoma, oligodendroglioma, non-tumor tissue). Three biopsies could not be classified due to non-specific DNA methylation results, and, for one biopsy, the classification was disregarded as it was incompatible with the IHC results for the same specimen.

### 2.6. Image Analysis

During surgery, the intraoperative US image space was fused to the imported PET/FLAIR, PET/ce-T1, FLAIR, and ce-T1 images in the navigation software. In Brainlab, the position of each image-localized biopsy was recorded in the US image space during sampling and converted to the other image spaces using the fusion transformation matrices. In SonoWand, US images and biopsy coordinates were recorded directly in the FLAIR MRI image space. Post-surgery, the intraoperative US images were rigidly registered to the FLAIR images using ImFusion software (version 2.18.0, ImFusion GmbH, Munich, Germany) to adjust for brain shift or fusion inaccuracy occurring during surgery. This process is illustrated in Appendix A. The registration was performed using a Linear Correlation of Linear Combination (LC2) algorithm, which has previously been shown to be suitable for US-MRI image registration for brain shift correction [24,25]. The post-registration transformation matrix for the US images was applied to the biopsy coordinates in the FLAIR image space to find brain shift-corrected coordinates. For 6 patients, intraoperative US was unavailable, and, for 2 patients, intraoperative US was conducted but brain shift correction could not be performed due to loss of US data (n = 1) or deformation of the tumor from the US probe (n = 1). For these patients, uncorrected biopsy coordinates registered directly in the FLAIR image space were used instead.

Quantitative image analysis was performed in PMOD. All ce-T1 MRI images and PET datasets were rigidly co-registered to the corresponding FLAIR images to ensure proper alignment between all datasets. To determine [^18^F]-FACBC uptake, a standardized volume of interest (VOI) with a radius of 2 mm was centered on each biopsy location. Tumor-to-background ratios (TBRs) were defined by dividing the mean standardized uptake value (SUV_mean_) in the VOI by the background uptake (SUV_background_). The background uptake region was defined in the contra-lateral side of the brain, above the ventricles, consisting of six consecutive, crescent-shaped regions of interest (ROIs), forming a VOI, as suggested by Unterrainer et al. [26].

MRI tumor volumes (FLAIR and ce-T1) were defined by a neuroradiologist and a physicist together. For each tumor, a large spherical VOI was placed manually to cover the whole tumor, and threshold values were adjusted to separately segment the high-intensity regions in the FLAIR images and the contrast-enhanced regions from ce-T1 MRI. The resulting VOIs were further manually adjusted to fit the visual volume. Biopsies were determined as positive for contrast enhancement if there was any overlap between the biopsy VOI and the ce-T1 volume. PET volumes were defined by applying the TBR threshold value to large spherical VOI placed to cover the MRI volumes and surrounding areas. Any signal situated outside the brain matter was then manually removed from the VOIs. Biopsies were determined as PET-positive if the average TBR value was higher than the threshold value.

### 2.7. Statistical Analysis

All statistical calculations were performed in RStudio. To compare differences in TBR between high-grade glioma, low-grade glioma, and control tissues, as well as between glioma types (astrocytoma, oligodendroglioma, glioblastoma, control tissue), a Kruskal–Wallis H test was performed. To compare TBR between low- and increased-cell-density tissues and IDH1 and ATRX mutation status, the Mann–Whitney U test was used. Receiver operator characteristics (ROCs) curve analyses were used to evaluate the accuracy of [^18^F]-FACBC for the classification of glioma grade, cell density, and IDH1 and ATRX mutation status. The following classification was used for area under the curve (AUC) discrimination: AUC < 0.5 was considered as none, 0.5 ≤ AUC < 0.7 as poor, 0.7 ≤ AUC < 0.8 as acceptable, 0.8 ≤ AUC < 0.9 as excellent, and ≥0.9 as outstanding [27]. Optimal threshold values were defined as the point on the ROC curve closest to the top lefthand corner. PET volumes determined using the resulting optimal threshold value were compared to the corresponding FLAIR and ce-T1 volumes using the Dice coefficient.

## 3. Results

### 3.1. Histomolecular Analysis

Based on IHC and DNA methylation analysis results, 48 image-localized biopsies were classified as HGG (n = 16), LGG (n = 25), or non-tumor tissue (n = 7). Tumor tissue samples were also classified as IDH1-wildtype or IDH1-mutated (n = 19 and n = 22, respectively), ATRX-retained or ATRX-lost (n = 19 and n = 13, respectively), and low- or increased-cell-density (n = 21 and n = 17, respectively). In total, 26 biopsies were additionally classified as glioblastoma (n = 7), astrocytoma (n = 13), or oligodendroglioma (n = 6) based on DNA methylation analysis. Full histopathological results for each biopsy are provided in Appendix A.

### 3.2. Quantitative Image Analysis

TBR, grade, diagnosis, and ce-T1 uptake for each biopsy are presented in Table 1. Median TBR was 2.0 (IQR 1.5–3.5) for tumor tissue samples and 1.7 (IQR 1.0–3.0) for control tissue samples. Median TBR for samples classified as low-grade and high-grade glioma was 1.8 (IQR 0.9–3.1) and 2.8 (IQR 1.7–6.4), respectively. No difference was found in TBR between tumor tissue (2.0, IQR 1.5–3.5) and non-tumor tissue (1.7, IQR 1.0–3.0) (*p* = 0.599), but TBR was significantly higher for HGG than non-HGG (1.8, IQR 0.9–3.1) (*p* = 0.037). The samples classified into types had median TBRs of 1.5 (IQR 0.8–1.9), 1.7 (IQR 1.5–6.0), and 2.7 (IQR 2.2–5.1) for astrocytoma, oligodendroglioma, and glioblastoma, respectively. No significant differences were found between glioma types (*p* = 0.092). TBRs for glioma grades and types are shown in Figure 3.

Figure 4 shows the differences in TBR for IDH1, ATRX, and cell density. TBR for tumor tissue classified as IDH1-wildtype was found to be significantly higher than for IDH1-mutated tumor tissue (*p* < 0.001). Likewise, tumor tissue with retained ATRX showed significantly higher TBRs compared to tumor tissue with ATRX loss (*p* < 0.010). Samples with increased cell density also had higher TBRs than samples designated as low-cell-density (*p* = 0.049).

### 3.3. ROC Analysis

The results from the ROC analysis for glioma grade, cell density, IDH1 status, and ATRX status are presented in Figure 5 and Table 1 together with the corresponding ce-T1 sensitivity and specificity. ROC analysis showed that the [^18^F]-FACBC uptake in terms of TBR had poor performance when classifying tumor tissue (AUC = 0.56, 95% CI: 0.33–0.79), high-grade tumor tissue (AUC = 0.69, 95% CI: 0.53–0.84), and increased cell density (AUC = 0.69, 95% CI: 0.52–0.86). However, TBR was acceptable for classifying ATRX-retained tumor tissue (AUC = 0.78, 95% CI: 0.63–0.93) and excellent for classifying IDH1-wildtype tumor tissue (AUC = 0.83, 96% CI: 0.7–0.95). The optimal threshold value for both of these responses was TBR = 2.0. At this threshold, TBR uptake predicted IDH1-wildtype tumor tissue with lower specificity than ce-T1 (0.76 vs. 0.79) but much higher sensitivity (0.88 vs. 0.41). Likewise, specificity was lower for ATRX-retained tumor tissue compared to ce-T1 (0.78 vs. 0.8), but sensitivity was much higher (0.74 vs. 0.42).

### 3.4. Volume Comparisons

To investigate if [^18^F]-FACBC uptake could be useful to identify regions of aggressive tumor tissue in IDH1-wildtype tumors, PET volumes for patients diagnosed with glioblastoma (n = 8) were defined by applying the optimal TBR threshold > 2 for IDH1-wildtype tumor tissue. The PET volumes were compared to the reference standard ce-T1 and FLAIR volumes. Table 2 shows the percentages of the PET volume outside and inside the ce-T1 and FLAIR volumes for each patient. PET volumes were on average over eight times larger than the ce-T1 volumes (PET vol/ce-T1 vol: 8.07 ± 7.29) and about half the size of the FLAIR volumes (PET vol/FLAIR vol: 0.56 ± 0.28). For the seven patients with ce-T1 uptake, the average percentage of the PET volume outside the ce-T1 volume was 80.9% ± 13.2% (range: 61.6–90.2%), while, for the FLAIR volume, it was 22.5% ± 18.8% (range: 0–56.7%).

In total, 19 biopsies were sampled from the eight patients with glioblastoma, of which 16 were classified as tumor tissue and 3 as non-tumor tissue. With a TBR threshold of 2, the PET volume included 14 of the tumor-positive biopsies (87.5%), while the ce-T1 volumes included 5 (31.3%). Of the three biopsies classified as non-tumor tissue, one fell within the PET volume, while two fell within the ce-T1 volumes. Examples of PET volumes and MRI volumes for three of the patients are shown in Figure 6.

## 4. Discussion

This is the first study to validate [^18^F]-FACBC uptake against histomolecular features in gliomas using image-localized biopsies. The key finding was that [^18^F]-FACBC detected glioma tissue more accurately than ce-T1 MRI, especially in patients with glioblastomas.

Gliomas have a heterogeneous nature, sometimes containing areas of different activity and molecular properties [3]. This heterogeneity presents a challenge in the diagnosis of gliomas, as the most aggressive regions of the tumor should be identified for accurate histopathological diagnosis. In our study, [^18^F]-FACBC uptake was highest in HGG and glioblastoma tissue. At the optimal threshold value TBR > 2, [^18^F]-FACBC uptake was excellent for discriminating IDH1-wildtype tumor tissue, synonymous with glioblastoma in this patient cohort. For contrast-enhanced tumors, the enhanced volume was assumed to correlate with areas of highly malignant tumor tissue. However, when evaluating only the glioblastoma patients in the current study, [^18^F]-FACBC PET detected 87.5% of tumor-positive biopsies, compared to only 31.3% for ce-T1, and, even though the PET volumes were on average eight times larger than the ce-T1 volumes, they included only one false positive compared to two for ce-T1. This indicates that [^18^F]-FACBC PET could be used with ce-T1 MRI to identify highly malignant tumor regions for biopsy sampling with higher accuracy.

Compared to the more established amino acid tracers such as [^11^C]-MET, [^18^F]-FET, and [^18^F]-FDOPA, [^18^F]-FACBC has demonstrated lower uptake in the normal brain parenchyma, yielding higher TBR values for high-grade tumors [9,11]. While glioblastoma patients displayed consistently high [^18^F]-FACBC uptake, most samples from the astrocytoma patients had low or no uptake of the tracer, and no statistically significant differences were found when comparing all tumor-positive biopsies to the non-tumor biopsies. These results align with previous studies that have also reported low uptake of [^18^F]-FACBC in low-grade gliomas [8,28], and suggests that PET imaging with other AA tracers with more consistent uptake in low-grade gliomas such as [^11^C]-MET, [^18^F]-FET, or [^18^F]-FDOPA could be more suitable for these patients.

In oligodendroglioma samples, uptake was inconsistent, with some low-grade tumor-positive biopsies displaying very high TBR values and some high-grade biopsies displaying no uptake. Previous studies using AA tracers [^18^F]-FDOPA and [^11^C]-MET have reported high uptake in oligodendrogliomas compared to other low-grade tumors, possibly reflecting unique amino acid metabolism properties for this tumor type [15,29]. This effect was also found in the current study, with high uptake for several of the low-grade oligodendroglioma samples, one of which displayed the second-highest TBR value of all the samples (TBR = 11.3). However, this was not seen consistently in this study as [^18^F]-FACBC uptake was highly varied in oligodendroglioma.

Threshold values for the delineation of PET volumes have previously been suggested in practice guidelines for AA tracers [^11^C]-MET, [^18^F]-FET, and [^18^F]-FDOPA [18], while this is the first study to evaluate a corresponding value for [^18^F]-FACBC based on histomolecular features. The Response Assessment in Neuro-Oncology (RANO) group now proposes using a consistent TBR threshold value of 1.6 for AA PET, regardless of which tracer is used [17]. However, as previously discussed, [^18^F]-FACBC uptake differs from the other AA tracers in certain glioma types and normal brain parenchyma, possibly due to differences in transport mechanisms [12,13]. Thus, threshold values suitable for other AA PET tracers may not necessarily be optimal for [^18^F]-FACBC. In this study, a TBR threshold value of 2 was determined as optimal for the delineation of IDH1-wildtype tumor tissue using [^18^F]-FACBC in patients with glioblastoma. As [^18^F]-FACBC has overall higher TBR values in high-grade tumor tissue than other AA tracers, using the suggested lower threshold of 1.6 might reduce the specificity in the delineation of these high-grade areas. However, [^18^F]-FACBC PET could not discriminate tumor tissue (including both LGG and HGG biopsies) from non-tumor tissue sufficiently well to determine an effective threshold value. This is likely due to the varied behavior of the tracer in low-grade tumors, as many of the included LGG biopsies did not display any uptake. While this suggests a decreased utility of [^18^F]-FACBC PET in LGG overall, the tracer could still provide useful information in the low-grade tumors that do demonstrate uptake. In these patients, using [^18^F]-FACBC PET at the higher threshold of TBR = 2 could still be beneficial to maintain a higher specificity for the detection of active tumor tissue.

Traditionally, the surgical target volume in IDH1-wildtype glioblastoma is based on the ce-T1 MRI volume, as numerous studies have shown that survival is associated with both the extent of surgical resection and residual ce-T1 volume [30,31]. More recently, several studies have argued for an additional effect of supramaximal surgical resection or FLAIR-ectomy, targeting non-enhancing tissue adjacent to the ce-T1 core, for tumors where such extensive resections can be performed safely. The RANO resect group found a better prognosis in patients without ce-T1 remnants and a residual FLAIR volume of less than 5 mL compared to patients who underwent maximal resection (0–1 cm^3^ ce-T1 remnants, >5 mL residual FLAIR volume), though there is no trial data yet, and selection bias may be an issue [32]. However, the literature reports conflicting results about the effect of supramaximal resections on recurrence and survival and how much of the FLAIR signal should be removed [33,34]. While infiltrating tumor cells in peri-tumoral FLAIR volumes likely contribute to disease recurrence, these regions contain far fewer tumor cells than the ce-T1 tissue [35], and, radiologically, it can be difficult to separate peritumoral vasogenic edema and non-enhancing tumor tissue from traditional MRI sequences [36]. The same challenge is present in determining target volumes for radiotherapy. In Europe, the ce-T1 MRI volume plus 1.5 cm margin is the clinical target volume (CTV) for radiotherapy [37], while the North American guidelines describe a two-phase approach targeting the whole FLAIR hyperintensity region followed by a boost to the resection cavity and gross tumor [38]. PET/RANO guidelines now propose that the delineation of a biological target volume (BTV) based on AA PET could allow for the improved treatment of high-risk tumor subvolumes not identified by MRI [39]. Potentially, [^18^F]-FACBC PET could be used to better identify areas of likely active tumor tissue outside the ce-T1 volume, with improved specificity compared to targeting the entirety of the FLAIR hyperintensity region, and could help put the line in the sand regarding how much of the FLAIR volume to target in both supramaximal surgical resections and/or radiotherapy.

### Limitations

One limitation of this study is the relatively small sample size. In particular, few patients with low-grade gliomas and less common glioma types were recruited, resulting in few biopsy samples in these groups compared to the more common glioblastomas. This could have impacted the robustness of the statistical analyses for the less common tumor types. Additionally, biopsies could only be sampled from areas of increased FLAIR signal for ethical reasons, meaning that any infiltrative tumor spread into areas that appeared normal on MRI may have been missed. This could have impacted the accuracy calculations for [^18^F]-FACBC PET. Volume comparisons in glioblastoma patients showed that while most of the PET volume was contained within the FLAIR volume, parts of the [^18^F]-FACBC uptake extended beyond the FLAIR hyperintensity regions for most patients. Without biopsies from mismatched PET-positive/FLAIR-negative areas, it is unknown whether [^18^F]-FACBC PET could detect infiltrative tumor foci beyond FLAIR volume borders or whether these areas represent false positives caused by the partial volume effect. This, in combination with the relatively small sample size, also resulted in few biopsies being defined as non-tumor tissue, which may have impacted calculations of specificity for [^18^F]-FACBC PET and ce-T1.

The accurate localization of biopsies is important for determining accurate TBRs and ce-T1-positive/negative status for each sample. In this study, intraoperative US was used to correct for uncertainty caused by brain shift during surgery. As this provided more precise localization of biopsy sites, smaller VOIs could be used and more representative average TBRs could be determined. However, small VOIs might also be prone to misregistration errors, particularly for patients where brain shift correction could not be performed due to intraoperative US being unavailable or inapplicable, resulting in increased uncertainty in the localization of biopsies from these patients.

## 5. Conclusions

[^18^F]-FACBC PET is more sensitive than ce-T1 MRI in detecting regions of viable glioma tissue, particularly in glioblastomas. The addition of [^18^F]-FACBC PET could be beneficial for identifying highly malignant tumor tissue for biopsy sampling and inform the delineation of target volumes for radiotherapy and surgical resection in patients with glioblastoma.

## Figures and Tables

**Figure 1 cancers-16-02581-f001:**
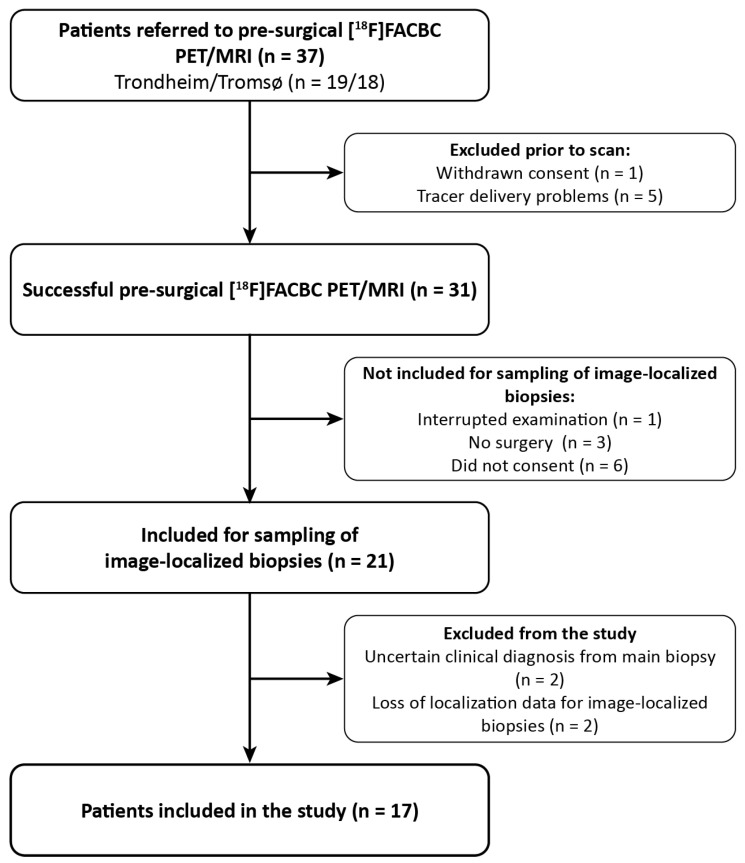
Patient inclusion flowchart.

**Figure 2 cancers-16-02581-f002:**
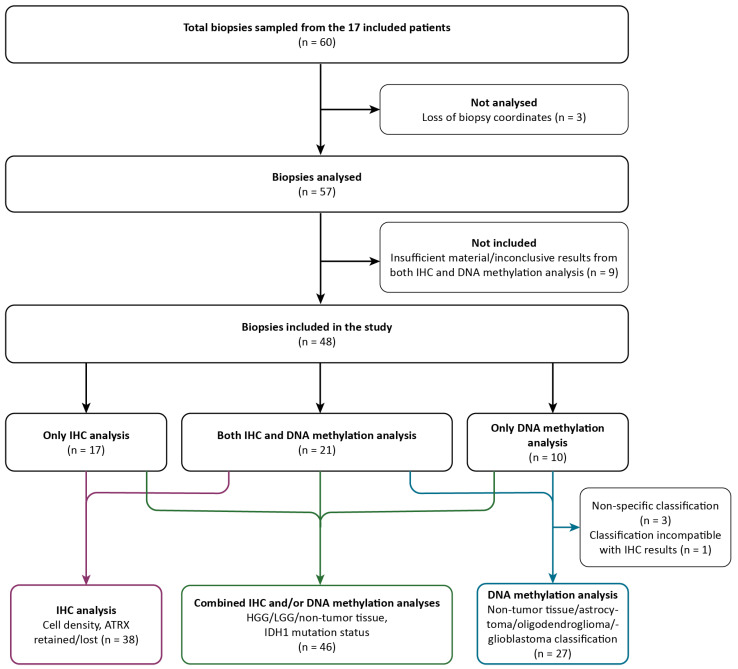
Flowchart of the inclusion process and classification of samples based on IHC analysis only (purple arrows), combined IHC and/or DNA methylation analysis (green arrows), and DNA methylation analysis only (blue arrows). IHC, immunohistochemical; LGG, low-grade glioma; HGG, high-grade glioma.

**Figure 3 cancers-16-02581-f003:**
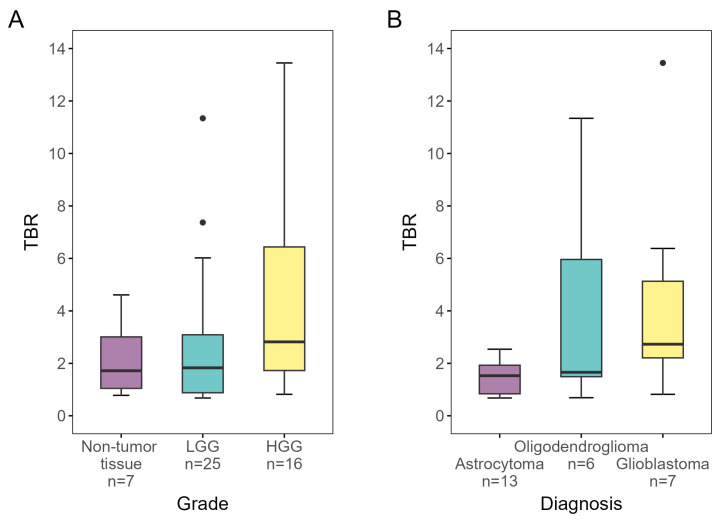
[^18^F]-FACBC TBRs for biopsies designated as (**A**) non-tumor tissue, low-grade and high-grade, and (**B**) astrocytoma, oligodendroglioma, and glioblastoma, as determined by IHC analysis and/or DNA methylation analysis. TBR was significantly higher for HGG than non-HGG (*p* = 0.037). LGG, low-grade glioma; HGG, high-grade glioma; TBR, tumor-to-background ratio.

**Figure 4 cancers-16-02581-f004:**
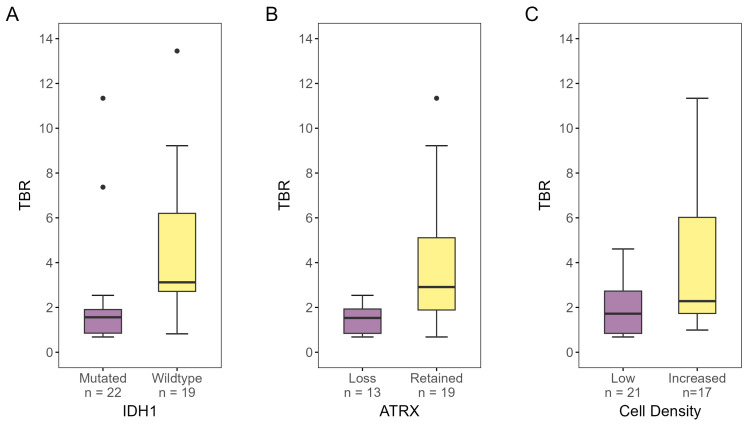
[^18^F]-FACBC TBR for biopsies designated as (**A**) IDH1-mutated or -wildtype, (**B**) ATRX-lost or -retained, and (**C**) low- or increased-cell-density, as determined by IHC analysis and/or DNA methylation analysis. There was a significant difference in TBR for all three properties (IDH1 status: *p* < 0.001, ATRX status: *p* < 0.01, cell density: *p* = 0.049). TBR, tumor-to-background ratio.

**Figure 5 cancers-16-02581-f005:**
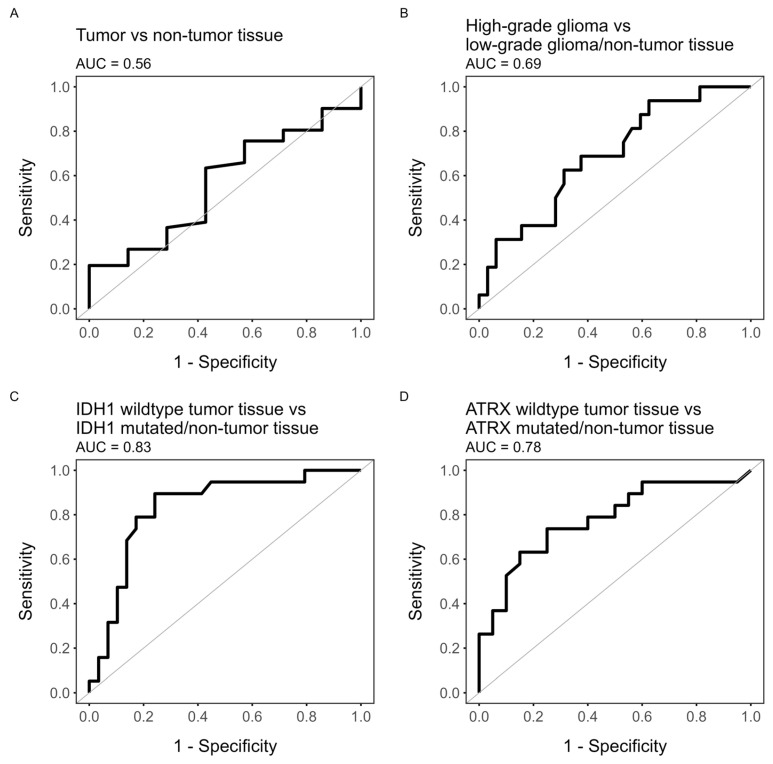
ROC plots of TBR value as predictor for (**A**) tumor tissue (HGG/LGG vs. non-tumor tissue), (**B**) high-grade glioma (HGG vs. LGG/non-tumor tissue), (**C**) IDH1-wildtype tumor (IDH1wt tumor tissue vs. IDH1-mutated tumor tissue/non-tumor tissue), or (**D**) ATRX-retained tumor (ATRX-retained tumor tissue vs. ATRX-lost tumor tissue/non-tumor tissue).

**Figure 6 cancers-16-02581-f006:**
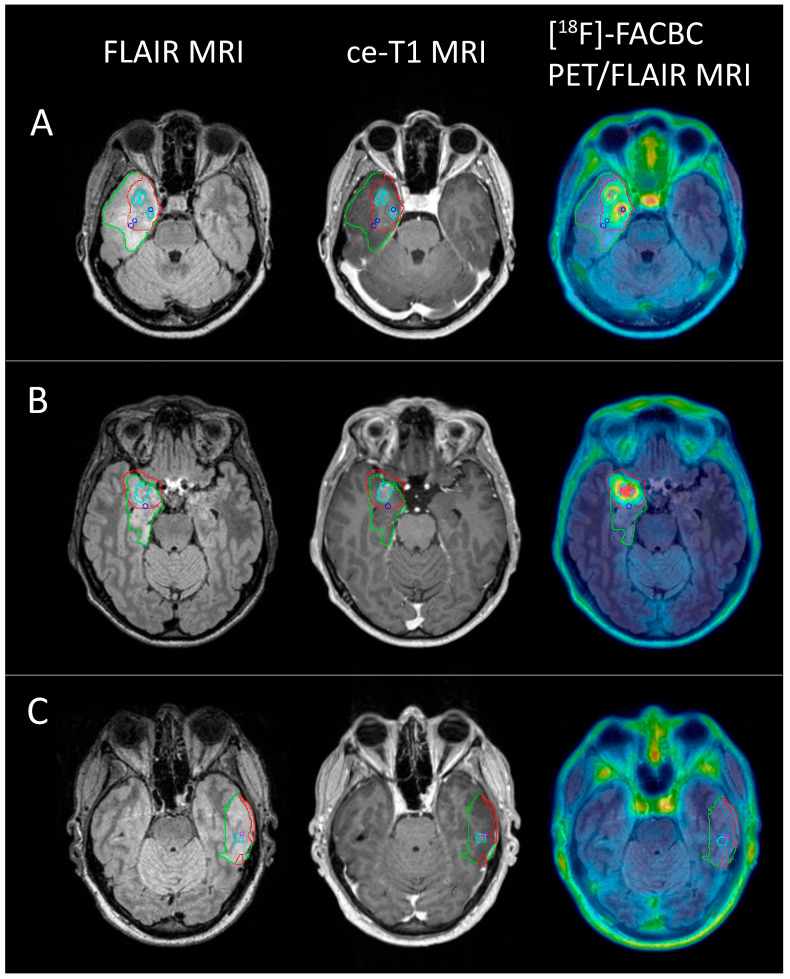
FLAIR (green), PET (red), and ce-T1 (cyan) MRI volumes for 3 glioblastoma patients ((**A**) female, 52; (**B**) male, 46; (**C**) female, 50) shown on FLAIR MRI, ce-T1 MRI, and PET/FLAIR images. Blue circles show biopsy coordinates classified as HGG (**A**,**B**), while the pink circle shows biopsy coordinates classified as non-tumor tissue (**C**). Two HGG biopsies from (**A**) and one from (**B**) did not show contrast enhancement, but were positive for [^18^F]-FACBC PET at a threshold of TBR = 2. One non-tumor biopsy from C was ce-T1 positive, but did not have [^18^F]-FACBC uptake.

**Table 1 cancers-16-02581-t001:** Results from ROC analysis: optimal threshold, sensitivity, specificity, AUC, 95% CI for [^18^F]-FACBC TBR for determination of tumor grade, increased cell density, and IDH1 and ATRX status, as well as corresponding ce-T1 sensitivity and specificity.

	TBR				Ce-T1	
	Threshold	Sensitivity	Specificity	AUC (95% CI)	Sensitivity	Specificity
Tumor tissue	1.73	0.63	0.57	0.56 (0.33–0.79)	0.24	0.43
High-grade glioma	2.13	0.69	0.62	0.69 (0.53–0.84)	0.25	0.69
Increased cell density	1.73	0.76	0.52	0.69 (0.52–0.86)	0.35	0.71
IDH1 wt tumor tissue	2.00	0.89	0.76	0.83 (0.71–0.96)	0.4	0.79
ATRX wt tumor tissue	2.00	0.74	0.75	0.78 (0.63–0.93)	0.42	0.80

**Table 2 cancers-16-02581-t002:** Comparisons of PET volumes (TBR > 2) and MRI FLAIR/ce-T1 volumes for patients diagnosed with glioblastoma.

Patient (Sex, Age)	Ce-T1 Volume (mL)	FLAIR Volume (mL)	PET Volume (mL)	% of PET Vol (TBR > 2) Outside Ce-T1 Vol	PET/Ce-T1 Dice Coefficient	% of PET Vol (TBR > 2) Outside FLAIR Vol	PET/FLAIR Dice Coefficient
Male, 47	0	16.1	13.3	100	0	10.3	0.81
Female, 52	1.5	57.3	15.5	90.2	0.18	0.0	0.43
Male, 71	27.1	130.0	66.7	61.6	0.55	25.7	0.50
Male, 46	2.0	26.3	19.5	90.0	0.18	31.3	0.59
Female, 50	0.3	40.5	6.9	97.0	0.06	0.0	0.29
Male, 74	1.6	11.6	7.2	81.4	0.30	27.5	0.56
Male, 80	7.3	36.4	34.7	80.8	0.32	56.7	0.42
Female, 55	13.2	74.7	25.8	65.4	0.46	28.2	0.37

## Data Availability

The datasets generated and/or analyzed during the current study are not publicly available due to the European Union General Data Protection Regulations (GDPR), but are available from the corresponding author upon reasonable request.

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
