# Peer review of "Histomolecular Validation of [18F]-FACBC in Gliomas Using Image-Localized Biopsies"

_cancers, 2024, doi:10.3390/cancers16142581_

Round 1

Reviewer 1 Report

Comments and Suggestions for Authors

"Histomolecular validation of [18F]-FACBC in gliomas using image-localized biopsies" is a prospective 2-center study of the correlation between pre-biopsy FACBC (fluciclovine) PET and biopsy results in patients with gliomas. Results suggest that there is high uptake in IDH1-wildtype gliomas and FACBC PET can fairly accurately identify and distinguish and delineate IDH1-wildtype gliomas from other processes and normal brain parenchyma.

The primary strength of the study is histopathologic correlation. AA PET is a very promising technique for initial imaging of high grade gliomas and having data to support its validity is critical for adoption of AA PET including FACBC in management of these patients.

Limitations:

- Small #patients, and exclusion of ~half of patients (only 17 of 31 patients who underwent PET were included)

- Sampling was based on both MR and PET, making it non uniform and confounding the results.

Minor issues:

Introduction, 3rd para: Need to mention fluciclovine which is a commonly used name in the literature for FACBC

It’s helpful to briefly state the mechanism of uptake (LAT system) and effect of BBB.

Introduction, line 62: “Additionally, higher tumor-to-background ratios (TBR) were found for [18F]-FACBC compared to the current recommended AA PET tracers [11C]-MET, [18F]-FET, and [18F]-FDOPA, due to lower uptake in normal brain tissue.” Needs reference. In general, I don't think there is evidence that FACBC is overall superior to FET or MET. FDOPA has physiologic uptake in basal ganglia so it might be inferior. 

Methods:

1- Static images were acquired 30-45 min p.i. Is this based on prior work? Typically, AA PET is a little earlier (e.g., 20-40 min) as they can washout. Please cite a reference or provide justification. The authors have list-mode data so they can potentially examine if the exact timing matters. Earlier static PET (10-30 min) might be more suitable for more aggressive tumors

2 - “a standardized volume-of-interest (VOI) with a radius of 2 mm was centered on each biopsy location” Please verify if 2-mm is correct. VOIs in the literature are typically larger, 1 mL volume if not more (e.g., EANM recommends 2 mL). I don’t know what the voxel size was (reconstructed PET FOV is not provided) but with a 4-mm Gaussian filter the effective resolution of PET is lower than the 2-mm VOI. The size of the VOI should be larger than expected error in misregistration of biopsy site.  Providing data or assumptions on misregistration error would be helpful.

3 - PET volume definition based on TBR threshold. This is not standard, and the optimal diagnostic threshold depends on prevalence so it is not fixed.

 Results: please provide data on biopsies from PET hot spots versus PET negative regions.

Table 2: No need to specify the arbitrary patient number (starting from 10 is also confusing). Instead, can specify age and sex, and/or location of the lesion. Also for patient 10, Ce-T1 volume should be stated as 0 (and 100% for PET outside enhancement) 

Reviewer 2 Report

Comments and Suggestions for Authors

I have reviewed the manuscript titled "Histomolecular validation of [18F]-FACBC in gliomas using image-localized biopsies." I congratulate the authors on a well-executed study on an interesting and emerging topic, on which my research group is also working, as evidenced by this review (PMID: 36612085). I would only suggest some minor revisions before accepting the work:

  1. Structure the abstract into introduction, materials and methods, results, and conclusions.
  2. Better specify and possibly add an image to explain how the biopsies were performed and how the brain shift was corrected, or how some biopsies were spatially considered unreliable. This seems to be the only weak point of the work, and I believe that providing extremely detailed information on how these biopsies were obtained would benefit the study.

The results are extremely convincing, and the discussion and conclusions are clear and well-written.
